# Promoting subjective preferences in simple economic choices during nap

Sizhi Ai[1,2†], Yunlu Yin[3,4†], Yu Chen[1], Cong Wang[5,6,7], Yan Sun[1], Xiangdong Tang[8], Lin Lu[1,5,7,9], Lusha Zhu[3,5,7,10]*, Jie Shi[1,11,12,13]*

[1]National Institute on Drug Dependence, Peking University, Beijing, China; [2]Department of Cardiology, Heart Center, Henan Key Laboratory of Neurorestoratology, The First Affiliated Hospital of Xinxiang Medical University, Weihui, China; [3]School of Psychological and Cognitive Sciences, Peking University, Beijing, China; [4]Faculty of Business and Economics, The University of Hong Kong, Hong Kong SAR, China; [5]Peking-Tsinghua Center for Life Sciences, Peking University, Beijing, China; [6]Academy for Advanced Interdisciplinary Studies, Peking University, Beijing, China; [7]IDG/McGovern Institute for Brain Research, Peking University, Beijing, China; [8]Sleep Medicine Center, State Key Laboratory of Biotherapy, West China Hospital, Sichuan University, Chengdu, China; [9]Institute of Mental Health, National Clinical Research Center for Mental Disorders, Key Laboratory of Mental Health and Peking University Sixth Hospital, Peking University, Beijing, China; [10]Key Laboratory of Machine Perception, Ministry of Education; Beijing Key Laboratory of Behavior and Mental Health, Peking University, Beijing, China; [11]Beijing Key Laboratory on Drug Dependence Research, Beijing, China; [12]The State Key Laboratory of Natural and Biomimetic Drugs, Beijing, China; [13]The Key Laboratory for Neuroscience of the Ministry of Education and Health, Peking University, Beijing, China

*For correspondence:
lushazhu@pku.edu.cn (LZ);
shijie@bjmu.edu.cn (JS)

†These authors contributed equally to this work

Competing interests: The authors declare that no competing interests exist.

**Abstract** Sleep is known to benefit consolidation of memories, especially those of motivational relevance. Yet, it remains largely unknown the extent to which sleep influences reward-associated behavior, in particular, whether and how sleep modulates reward evaluation that critically underlies value-based decisions. Here, we show that neural processing during sleep can selectively bias preferences in simple economic choices when the sleeper is stimulated by covert, reward-associated cues. Specifically, presenting the spoken name of a familiar, valued snack item during midday nap significantly improves the preference for that item relative to items not externally cued. The cueing-specific preference enhancement is sleep-dependent and can be predicted by cue-induced neurophysiological signals at the subject and item level. Computational modeling further suggests that sleep cueing accelerates evidence accumulation for cued options during the post-sleep choice process in a manner consistent with the preference shift. These findings suggest that neurocognitive processing during sleep contributes to the fine-tuning of subjective preferences in a flexible, selective manner.
DOI: https://doi.org/10.7554/eLife.40583.001

## Introduction

Sleep complements wakefulness by supporting an array of cognitive functions. Yet its role in complex behavior such as value-based decision-making remains to be explored. Substantial evidence suggests that the storage of reward-related information does not stay sedentary during sleep. Spontaneous neural activation has been observed in the ventral striatum (***Peigneux et al., 2004***;

*Pennartz et al., 2004*; *Lansink et al., 2008*; *Lansink et al., 2009*) and other brain structures implicated in reward processing (*Cantero et al., 2003*; *Schabus et al., 2007*; *Singer and Frank, 2009*; *Fujisawa and Buzsáki, 2011*). This engagement of reward circuits has been postulated to promote the consolidation of reward-related information (*Pennartz et al., 2004*; *Fischer and Born, 2009*; *Pennartz et al., 2011*), and prioritize information of high motivational saliency for sleep-dependent reprocessing (*Fischer and Born, 2009*; *Sterpenich et al., 2009*; *Wilhelm et al., 2011*). Some studies also implicate connections of the active reward system in the sleeping brain with abnormal behavior, such as compulsive eating disorders, which have been associated with the dysfunction in reward networks (*Perogamvros et al., 2012a*). It remains to be explored, however, whether neural activity during sleep directly supports or modulates cognitive processes related to reward processing in ways that affect goal-directed behaviors, despite potential implications of such research for understanding the nature of mechanisms underlying sleep and reward processing and for clinical interventions aiming at modifying human behavior, such as overeating or smoking, to prevent diseases.

A classic model for research in the field of reward processing and value-based behavior entails preference-guided decisions (*Rangel et al., 2008*). Choosing among different options involves computing a subjective value for each option and comparing these values to make a decision (*Samuelson, 1938*; *Rangel and Hare, 2010*). The capacity to effectively evaluate resources and choices is critical for survival and thriving, and the ways of fine-tuning such valuation to reflect subtle changes in the environment or the cognitive state of a decision-maker represent important adaptive systems in animals. Indeed, extant data have demonstrated that subjective preferences are not formed once and forever, but can be modulated by external (e.g. reward reinforcement (*Tobler et al., 2005*)) and internal factors in the waking state. For example, memory retrieval has been shown to influence the weighting of choice options in certain types of decisions (*Barron et al., 2013*; *Gluth et al., 2015*; *Bornstein and Norman, 2017*), whereas attention affects the comparison of choice values in a range of economic behaviors (*Armel et al., 2008*; *Krajbich et al., 2010*; *Schonberg et al., 2014*). These data thus argue for labile preferences and provide a foundation for investigating the vulnerability of preferences to modulation during sleep.

Here, we explored the extent to which sleep-dependent neurocognitive processing affects the evaluation of intrinsic preferences that guide decisions while awake. In particular, we investigated whether preferences for choice options can be specifically targeted and modified during midday nap, using targeted memory reactivation (TMR), a procedure that has been applied to manipulate neurocognitive processing via stimulating the sleeping brain with unobtrusive cues. The method has been shown effective in selectively improving post-sleep performances in a range of cognitive tasks, including spatial memory (*Rasch et al., 2007*; *Rudoy et al., 2009*; *Diekelmann et al., 2011*), motor memory (*Antony et al., 2012*), vocabulary learning (*Schreiner and Rasch, 2015*; *Schreiner and Rasch, 2017*; *Tamminen et al., 2017*), fear extinction (*Hauner et al., 2013*; *He et al., 2015*), and the reduction of implicit social biases (*Hu et al., 2015*), but its influence on value-based decisions has yet to be explored.

Specifically, we investigated the effect of covert, reward-related stimulation to the sleeping brain on preferences and choices in an important class of behavior characterized by simple economic choices (e.g. choosing between apples and oranges). Such behavior is prevalent in human and non-human animals and has served as a major building block for understanding complex goal-directed behaviors (*Fehr and Rangel, 2011*; *Padoa-Schioppa, 2011*). Focusing on simple choices also allows us to exploit existing knowledge regarding cognitive mechanisms associated with decision-making, and in particular the known relationship between choice and reaction time and the distinct latent aspects of the decision process characterized by computational frameworks such as sequential sampling (*Smith and Ratcliff, 2004*; *Gold and Shadlen, 2007*). This framework thus allows us to explore how stimulating the sleeping brain affects the underlying cognitive processes that give rise to preferences and decisions.

We first examined whether TMR modifies behavior and whether such modification is sleep-dependent. We implemented TMR through presenting the spoken name of familiar snack items repeatedly and unobtrusively during non-rapid-eye-movement (NREM) sleep and found a significant post-sleep preference enhancement for items that had been cued relative to items not externally cued. In stark contrast, we found no significant differences in preferences and choices in a control group who underwent the same treatment but were exposed to verbal stimuli while awake. We then examined the extent to which neurocognitive processing during sleep contributes to post-sleep preference

shift. By monitoring brain activity during sleep using electroencephalographic recordings and relating this neural measure to behavioral evidence of preference shifts, we found that cueing-specific preference improvements can be reliably predicted by delta or theta band signals induced by cues during sleep. Finally, we fit the decision and reaction time data observed in simple choices using a well-established sequential sampling model to explore the influence of verbal cueing during sleep on the subsequent decision process.

## Results

We tested 92 healthy adults (Materials and methods and *Supplementary file 1*, Table 1A for demographic information) in a battery of classic simple economic choice experiments in combination with the TMR procedure (*Figure 1*). Subjects were first introduced to a set of 60 familiar snack items (self-reported familiarity rating = 3.51 ± 0.84, on a 5-point scale, *Supplementary file 1*, Table 1B), and indicated their willingness-to-pay (WTP) in a Becker-DeGroot-Marschak (BDM) auction (*Becker et al., 1964*), which served as a measure of the baseline preference for a snack item (henceforth WTP1) (*Plassmann et al., 2007*; *Hare et al., 2008*; *Schonberg et al., 2014*). To establish an association between snacks and verbal cues, the spoken name of each snack (e.g. 'M and M's candies') was delivered over a speaker along with the presentation of the item image on a computer screen during the BDM auction. On the basis of WTP1, we chose eight pairs of snacks for each subject that were matched by baseline preferences and randomly divided into cued and uncued conditions.

During subsequent sleep (sleep group, N = 47), the names of eight snack items (i.e. cued items) were delivered over a speaker, each for ten repetitions, during the stage 2 of non-rapid eye movement (N2) sleep with no sleep disruption (*Schreiner and Rasch, 2015*; *Cairney et al., 2018*). After waking, we probed the effect of verbal cueing on preferences by re-evaluating the WTP for all 60 items using the same BDM auction (henceforth WTP2). We also probed changes in choices by performing a binary decision task where subjects needed to select between a pair of snacks that were matched for the baseline WTP but differed by whether they had or had not been named during sleep. As a control, we also measured behavior in a control group where participants underwent the same experimental procedure but, rather than taking a 90 min nap as in the asleep condition, were kept awake for the same duration of time (wake group, N = 45; Materials and methods and *Supplementary file 1*, Table 1A). Sleep monitoring and staging were performed using standard criteria from polysomnography recordings (Methods).

### Verbal cueing during sleep but not wakefulness improves preferences for previously cued items

First, we investigated the effect of verbal cueing on the subjective value of a snack by comparing the paired difference of WTP elicited from the first and second BDM auction (i.e. WTP2-WTP1), which for simplicity we refer to as 'ΔWTP'. Consistent with previous evidence that WTP reliably reflects the subjective value of a familiar option (*Plassmann et al., 2007*), the baseline ΔWTP (i.e. ΔWTP for items that had not been cued) was not significantly different from zero in either subject group (all subjects, ΔWTP = 0.11 ± 0.12, $t_{91}$ = 0.99, p = 0.323; sleep vs. wake, $t_{90}$ = - 0.97, p = 0.336), suggesting the test-and-retest stability of preferences for snack items examined in our experiment.

Using ΔWTP, we found that there was no significant main effect in ΔWTP between the sleep and wake groups (sleep: 0.29 ± 0.12; wake: 0.32 ± 0.11; main effect in cohort, $F_{1,90}$ = 0.02, p = 0.886; *Figure 2A*). On the other hand, there was a significant within-subject difference associated with cueing, such that previously cued items were associated with higher ΔWTP relative to uncued items (cued: 0.50 ± 0.11; uncued: 0.11 ± 0.12; main effect in cueing condition, $F_{1,90}$ = 27.61, p = $9.9 \times 10^{-7}$; *Figure 2A*). Importantly, cue-induced effects on preferences varied between sleep and wake cohorts as predicted: ΔWTP was significantly higher for cued relative to uncued items in the sleep group but not in the wake group ((cued vs. uncued) × (sleep vs. wake): $F_{1,90}$ = 6.95, p = 0.01; *Figure 2A*), suggesting that verbal cueing during sleep but not wakefulness promotes preferences for previously cued items. This interaction was significant when analyzed using permutation tests that do not require specific distributional assumptions (*Figure 2—figure supplement 1*). The result was also robust after taking into account of filler items (*Figure 2—figure supplement 2*),

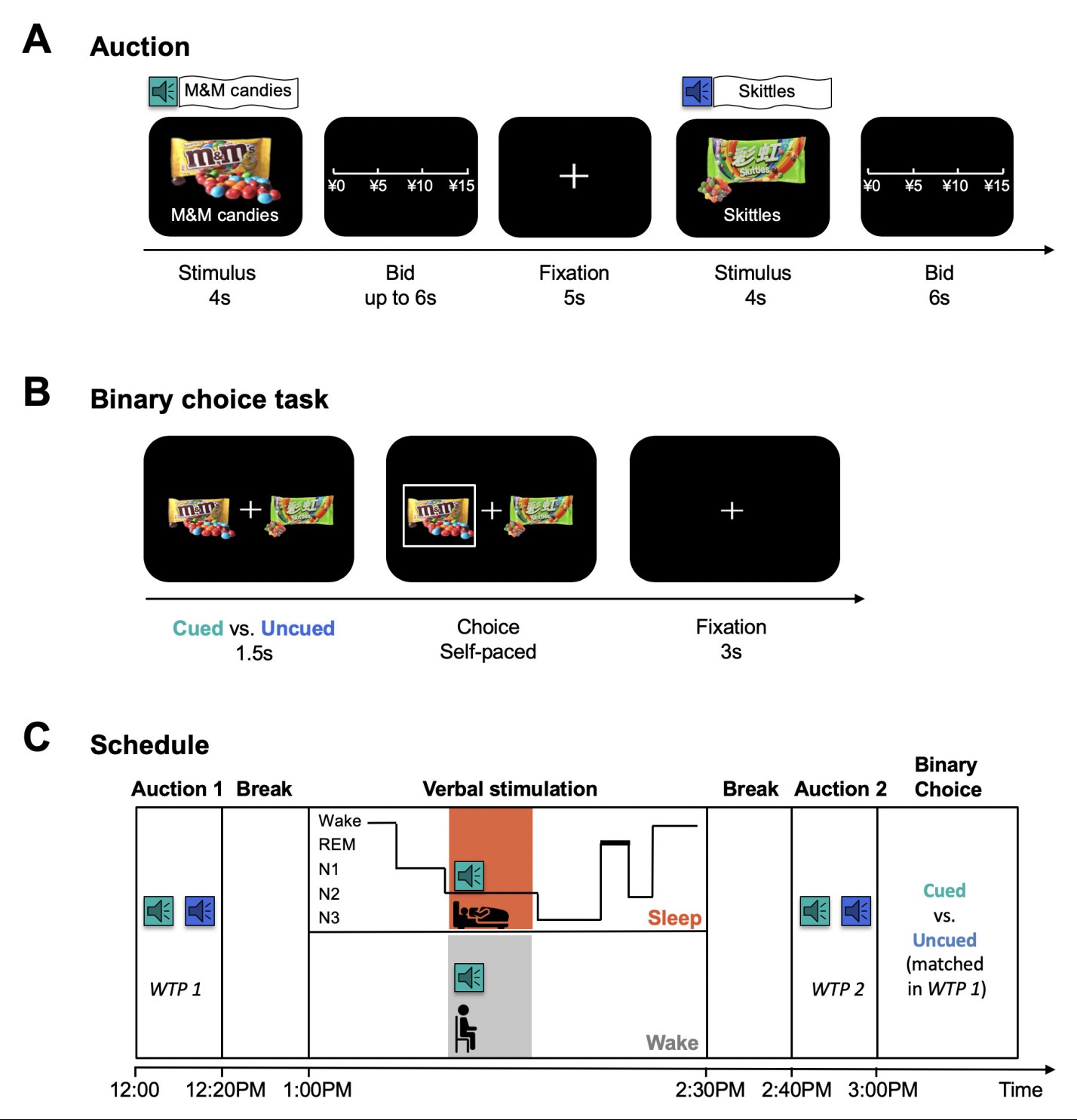

**Figure 1.** Task Illustration. (**A**) BDM auction. In each trial, participants were presented with a familiar snack item on a computer screen for 4 s, accompanied by the spoken name of the item delivered over a speaker once. Then, participants were asked to report their willingness to pay for the item. (**B**) Binary choice task. Participants were presented with two snack items that were matched by baseline WTP (i.e. WTP1) and differed by whether they had or had not been named during verbal stimulation. (**C**) Timeline of the experiment. Subjects first indicated their baseline WTP (i.e. WTP1) for a set of 60 familiar snacks in the first BDM auction. After a break, subjects entered a verbal stimulation session, during which they were randomly divided into the sleep (N = 47) or wake (N = 45) group. Sleep group participants took a 90-min nap. The names of 8 snacks were broadcast during N2 sleep, each with ten repetitions. Wake group participants received the same stimulation for the same number of snacks at approximately the same time that verbal cues were delivered to the sleep group. After a break, all subjects underwent a second BDM auction followed by a binary choice task.

DOI: https://doi.org/10.7554/eLife.40583.002

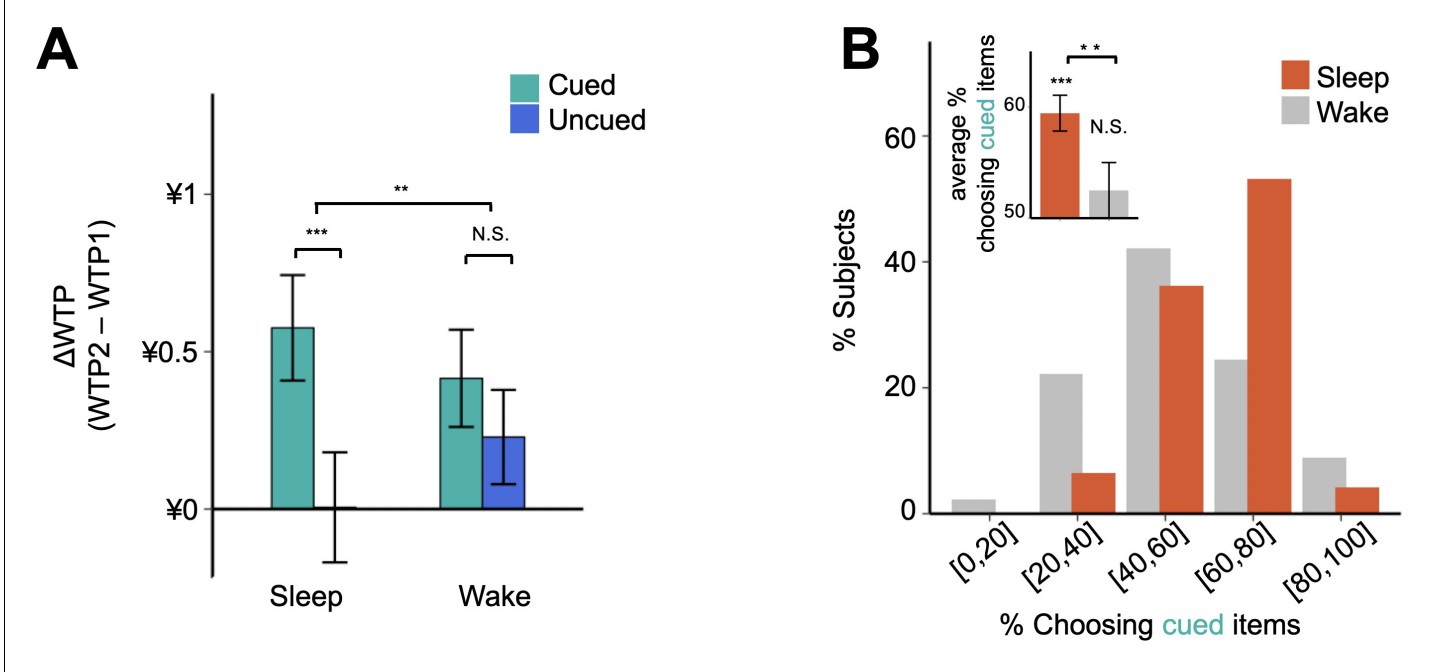

**Figure 2.** Effects of verbal cueing on preferences and choices. (A) Effects of cueing on WTP. ΔWTP is defined as the paired difference of an item's WTP elicited by the first and second BDM auction (i.e. WTP2 – WTP1). Relative to WTP1 (¥7.02 ± ¥0.16), cueing during sleep promotes preferences by 8.3% (±1.3%) for cued items more than for items not externally cued, whereas during wakefulness cueing increases WTP by 3.5% (±2.1%) for cued relative to uncued items. (B) Effects of cueing on choices in the binary decision task. Histograms illustrate the percentage of participants for each bin of choice likelihood. The average frequency of selecting the cued item was higher in the sleep group than in the wake group (Insert). Error bars indicate S.E.M.
DOI: https://doi.org/10.7554/eLife.40583.003

The following figure supplements are available for figure 2:

**Figure supplement 1.** Results of permutation test for the effect of verbal cueing on ΔWTP.
DOI: https://doi.org/10.7554/eLife.40583.004

**Figure supplement 2.** Detailed effects of verbal cueing on ΔWTP.
DOI: https://doi.org/10.7554/eLife.40583.005

demographic variables, and changes in self-reported measures of hunger and vigilance before and after the verbal cueing session (*Supplementary file 1*, Table 1C).

## Verbal cueing during sleep but not wakefulness biases choices toward previously cued items

To examine the extent to which the effect of verbal cueing translated into subsequent choice behavior, we next examined behavioral patterns in the binary decision task, where subjects needed to select within a pair of a cued and uncued items with similar WTP1. If, as suggested by the above results, cueing during sleep is associated with cueing-specific enhancement of preferences, we should expect a stronger tendency of choosing cued over uncued items in the sleep group. Consistent with this prediction, we found that the choice distribution of the sleep group significantly skewed toward cued items (Wilcoxon rank-sum test, $Z = -4.45$, $p = 8.4 \times 10^{-6}$; *Figure 2B*), indicating a behavioral bias favoring items previously named during sleep. In stark contrast, choices of the wake group were distributed almost symmetrically around the chance level (Wilcoxon rank-sum test, $Z = -0.88$, $p = 0.379$), indicating that the same verbal stimulation during wakefulness did not alter preferences for cued over uncued items. On average, sleep group subjects were more likely to select items that were previously cued than wake group participants (sleep: 59.4 ± 1.6%; wake: 52.5 ± 2.5%; Wilcoxon rank-sum test, $Z = 2.63$, $p = 0.009$; *Figure 2B* and *Supplementary file 1*, Table 1C).

## Verbal cueing during sleep does not influence choice randomness at either the subject or item level

To evaluate the possibility that, rather than preference shifts, the observed choice difference between the sleep and wake groups could be attributed to alternative factors such as changes in choice randomness, we compared the extent to which decisions were guided by value differences between the two participant groups. Specifically, we regressed the likelihood that subjects chose cued over uncued items against the difference in WTP2 (i.e. $WTP2_{cued} - WTP2_{uncued}$; *Figure 3A*). Consistent with previous studies (*Sugrue et al., 2005*), WTP2 differences predicted the choice behavior in both sleep and wake groups (sleep: $r = 0.37$, $p = 0.011$; wake: $r = 0.39$, $p = 0.009$), whereas WTP1 differences had no predictive power for the choice behavior in either subject group (sleep: $r = 0.01$, $p = 0.945$; wake: $r = 0.05$, $p = 0.746$). Importantly, if changes in choice randomness contribute to the observed behavioral bias under the sleep protocol, we would expect a steeper regression slope in sleep group participants relative to that of the wake group. In contrast to this prediction, the regression results showed no significant differences in slopes between the sleep and

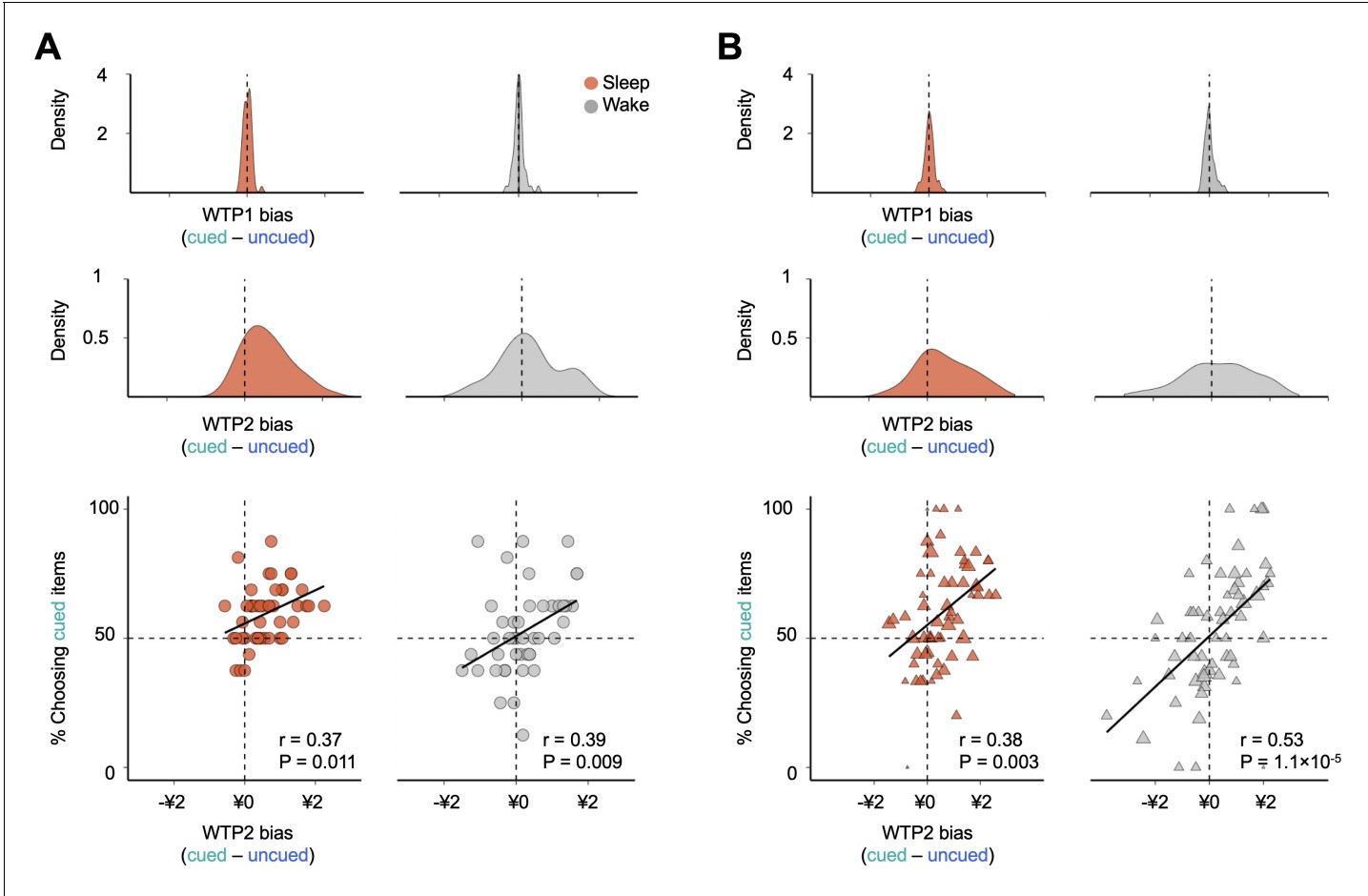

**Figure 3.** Effects of verbal cueing at the subject and item levels. (**A**) Density plot of individual-level biases in WTP1 (top) and WTP2 (middle) in the binary decision task. The individual WTP1(2) bias is computed by subtracting WTP1(2) of an uncued item from the competing cued items that a participant chose between in the binary decision task. (Bottom) Individuals who demonstrated stronger biases in WTP2 toward cued items were more likely to choose cued over uncued items in both sleep and wake groups, with no significant group difference between slopes, suggesting that the cueing-specific effect was not driven by differential level of choice randomness across sleep and wake groups. Each dot represents a subject. (**B**) Density plot of item-level biases in WTP1 (top) and WTP2 (middle). The item-level WTP1(2) bias is computed by averaging the WTP1(2) difference associated with the same cued item across all subjects and all uncued items paired in the binary decision task. (Bottom) Cued items that were associated with higher WTP2 biases were more likely to be selected under both sleep and wake protocols, with no significant group differences between slopes. Each triangle represents a cued item, and the size of the triangle is proportional to the number of observations.
DOI: https://doi.org/10.7554/eLife.40583.006

wake cohorts (sleep = 0.06; wake = 0.08; $F_{1,88}$ = 0.19, p = 0.665). Differences were found only in the intercepts (sleep = 0.56, wake = 0.51, $F_{1,88}$ = 6.34, p = 0.014; *Figure 3A*), suggesting that the behavioral differences between sleep and wake protocols could not be attributed to differential choice randomness but rather to an overall shift in choice propensity toward cued items under the sleep protocol. Similar results were obtained when examining the relationship between choices and WTP biases at the item level (*Figure 3B*).

## Cue-induced delta and theta band power predicts a post-sleep preference shift at both the subject and item level

Next, we investigated how the sleeping brain processed the spoken name of a snack and how such processing contributed to post-sleep behavior using event-related potential (ERP) data measured in a subset of sleep group participants (N = 23; *Supplementary file 1*, Table 1D, Table 1E; Materials and methods). By averaging ERP amplitudes measured over the interval from −200 to 1800 ms with respect to the onset of each verbal cue, we observed a cue-evoked neural response at the frontal electrode sites (representative electrode F3) during N2 sleep (*Figure 4A*). Characterized by a positive ERP around 450 ms followed by a negative peak around 1000 ms, this observed pattern is similar to the well-established waveform of K-complexes (KCs) (*Figure 4—figure supplement 1A* and *Supplementary file 1*, Table 1F) (*Niiyama et al., 1995*; *Cote et al., 1999*; *Blume et al., 2017*), an electroencephalogram (EEG) graphoelement that has been repeatedly associated with the processing of sensory stimuli, including auditory cues, mainly during periods identified as N2 (*Halász, 2005*; *Cash et al., 2009*).

We then examined how verbal cueing evoked power changes at different frequency bands by analyzing event-related changes in spectral power (ERSPs) before and after the cue onset. For each stimulus presentation, we segmented trials from −800 to 1800 ms with respect to the cue onset (*Ruch et al., 2014*; *Schreiner and Rasch, 2015*). Using the mean spectral power measured over the interval from −800 to −200 ms as a baseline, we computed the event-related change relative to the baseline from 0 to 1800 ms (*Schreiner and Rasch, 2015*). We observed no significant increase in high-frequency activity (8–35 Hz), which would have been observed if verbal stimuli had reduced the depth of sleep in participants (*Figure 4A*) (*Ruch et al., 2014*). More specifically, by contrasting the mean spectral power for high-frequency bands before and after the stimulus presentation, we found no significant change for the alpha, beta, or gamma bands (alpha: $t_{22}$ = - 1.51, p = 0.728; beta: $t_{22}$ = - 1.68, p = 0.535; gamma: $t_{22}$ = - 0.23, p = 1; all Bonferroni corrected). In sharp contrast, presentation of verbal stimuli elicited strong event-related synchronization for low-frequency activity (0.5–8 Hz), including the delta band ($t_{22}$ = - 6.88, p = $3.3 \times 10^{-6}$) (*Figure 4—figure supplement 1B*), which has been previously implicated in the detection of motivationally salient stimuli (*Blume et al., 2017*) and improved memory consolidation (*Oudiette et al., 2013*; *Batterink et al., 2016*), and the theta band ($t_{22}$ = - 7.13, p = $1.9 \times 10^{-6}$), which has been associated with memory processing during wakefulness (*Lisman and Jensen, 2013*) and successful cueing during non-rapid eye movement sleep in both healthy (*Schreiner and Rasch, 2015*) and patient (*Hot et al., 2011*) populations.

Importantly, the cue-induced change in power of the delta and theta bands strongly predicted the degree of cueing-specific preference enhancement after waking, at both the individual and item level. That is, among the sleep group subjects, individuals associated with more prominent increases in average delta or theta band power later demonstrated more pronounced post-sleep preference enhancements for cued items (delta: r = 0.65, p = 0.005; theta: r = 0.64, p = 0.004; all Bonferroni corrected; *Figure 4B–C*). Similarly, among items that had been presented during sleep, those associated with more prominent increases in average delta or theta band power showed greater post-sleep improvement in WTP (delta: r = 0.52, p = 0.001; theta: r = 0.47, p = 0.005; all Bonferroni corrected; *Figure 4D–E*). In contrast, there was no correlation between the post-sleep change in preference for cued items and the changes in power in the alpha, beta, or gamma bands (alpha: p = 0.463; beta: p = 1; gamma: p = 1; Bonferroni corrected; *Supplementary file 1*, Table 1G). Similarly, there was no correlation between the preference change for uncued items and the power change in either the high or low-frequency bands at either the subject or item level (alpha: p = 0.229; beta: p = 0.877; gamma: p = 1; Bonferroni corrected). These results suggest that preference modification following sleep TMR results from neurocognitive processing of verbal stimuli during sleep, as indexed by the power of low-frequency bands evoked by verbal cues.

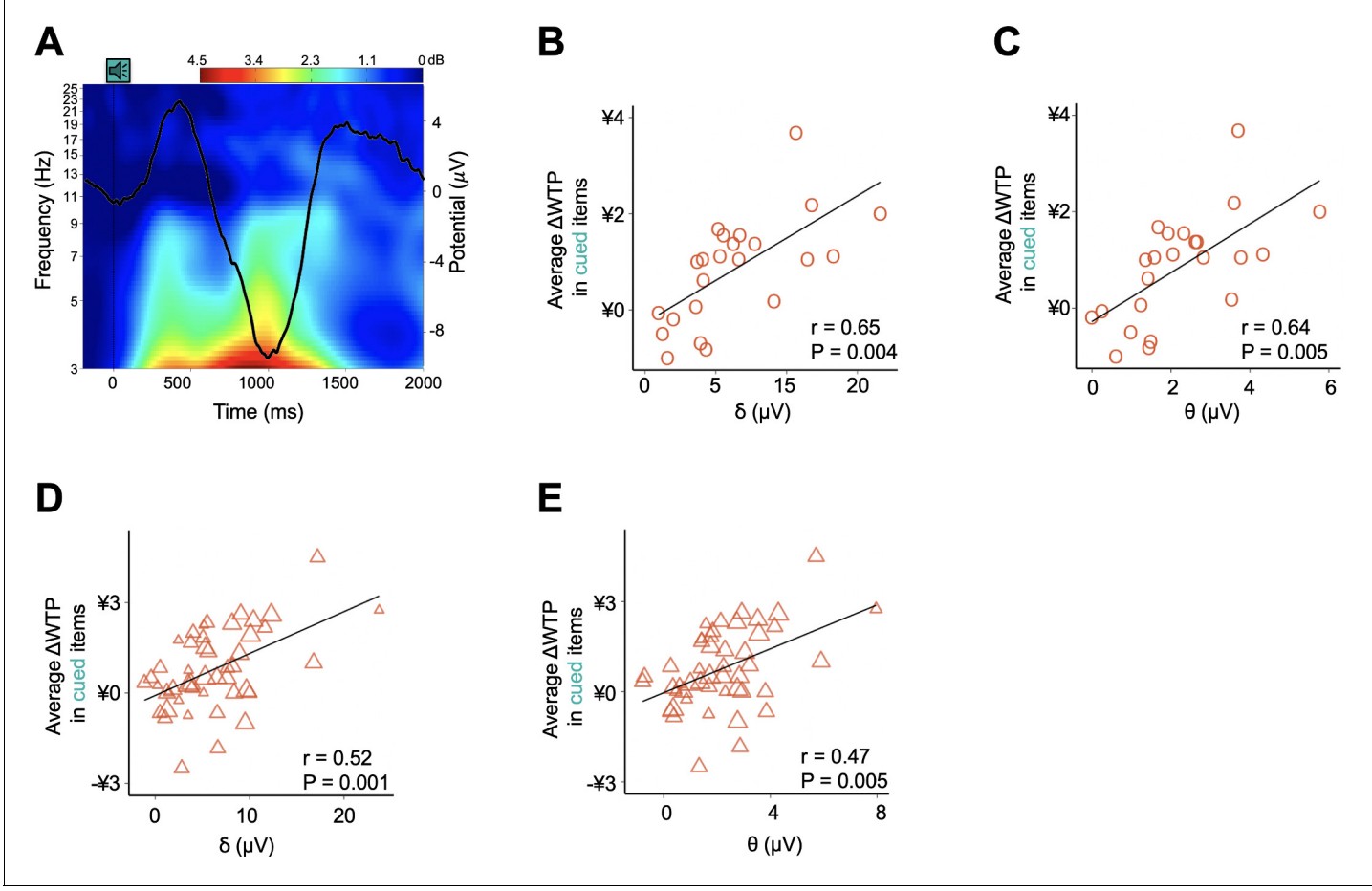

**Figure 4.** Neurophysiological responses to verbal stimuli during sleep and their contributions to post-sleep behavior (N = 23). (**A**) Event-related potential (ERP) and event-related changes in spectral power (ERSPs) evoked by the onset of verbal cues. EEG analysis only encompasses 23 participants in the sleep group. The heatmap demonstrates the event-related changes in ERSPs and black line demonstrates the grand average ERP across participants and trials at the representative electrode F3. ERP topographies for other electrodes and the detailed statistical analyses are shown in *Figure 4—figure supplement 1A* and *Supplementary file 1*, Table 1F (also see Materials and methods). (**B–C**) Individual- and (**D–E**) item-level correlations between cue-induced power increased in low-frequency bands and ΔWTP for items presented during sleep.
DOI: https://doi.org/10.7554/eLife.40583.007

The following figure supplements are available for figure 4:

**Figure supplement 1.** Neurophysiological responses to verbal cueing during sleep (N = 23).
DOI: https://doi.org/10.7554/eLife.40583.008

**Figure supplement 2.** Linear ballistic accumulation model fit.
DOI: https://doi.org/10.7554/eLife.40583.009

**Figure supplement 3.** Bar plots for estimated LBA parameters of the sleep and wake group.
DOI: https://doi.org/10.7554/eLife.40583.010

**Figure supplement 4.** Across-subject differences in LBA estimates.
DOI: https://doi.org/10.7554/eLife.40583.011

**Figure supplement 5.** Posterior predictive simulation revealed that the between-subject variability in the estimated drift rates was related to the variation observed in RT distributions.
DOI: https://doi.org/10.7554/eLife.40583.012

## Cueing during sleep selectively accelerates evidence accumulation for cued options

Finally, we explored how verbal cueing during sleep influenced the subsequent choice process by fitting choice and reaction time (RT) data with a well-established sequential sampling model, the Linear Ballistic Accumulation (LBA) model (*Brown and Heathcote, 2008*). The model has been applied to

study a wide range of decisions (*Forstmann et al., 2008*; *Forstmann et al., 2010*; *Hawkins et al., 2016*), including simple economic choices in a variety of forms (*Trueblood et al., 2014*; *Rodriguez et al., 2015*). At the heart of LBA is the idea that a decision arises from the accumulation of evidence for each possible option, and that a decision is made when an accumulator for one particular option reaches a predetermined threshold. This model has three key advantages that facilitate making inferences about cognitive components that underlie the behavioral change in the sleep group. First, it allows us to go beyond choice behavior and explore the influence of verbal cueing based additionally on the data of reaction time (RT) observed in binary decisions. Second, by assuming independent evidence accumulators for cued and uncued items respectively, it allows us to separately identify and directly compare the sequential sampling processes for items that have and have not been previously cued within each subject. Third, the model also allows us to examine if sleep exerts any general influence on the latent cognitive components that are either directly related to or separated from the valuation process.

Following prior studies (*Brown and Heathcote, 2008*; *Forstmann et al., 2008*; *Forstmann et al., 2010*), our LBA included five individual-level parameters: two drift rates for cued and uncued items respectively, starting point variability, decision threshold, and non-decision time (Materials and methods). Results based on a hierarchical Bayesian model estimation (*Gelman et al., 2014*) suggested that LBA explained RT and choice behavior very well: the observed choices and RT were highly correlated with predictions based on model estimation, with both correlation coefficients close to 1 (RT: $r = 0.995$, $p = 2.2 \times 10^{-16}$; choices: $r = 0.984$, $p = 2.2 \times 10^{-16}$; *Figure 4—figure supplement 2*) and no significant difference in the explanatory power between sleep and wake groups (RT: $r_{sleep} = 0.992$, $r_{wake} = 0.997$, $F_{1,88} = 0.42$, $p = 0.52$; choices: $r_{sleep} = 0.973$, $r_{wake} = 0.988$, $F_{1,88} = 0.50$, $p = 0.480$).

Having assessed model fit, we next examined whether drift rate estimates recovered from choice and RT in binary decisions would reproduce findings of WTP elicited from the separate auction task. Consistent with the ΔWTP pattern shown in *Figure 2A*, we found no significant main effect in the drift rate between sleep and wake groups (sleep = $3.17 \pm 0.06$; wake = $3.17 \pm 0.07$; main effect in treatment, $F_{1,90} = 0$, $p = 1$; *Figure 4—figure supplement 3A*), yet a significant within-subject difference between cued and uncued items, such that previously cued items were associated with higher drift rates (cued = $3.37 \pm 0.06$; uncued = $2.96 \pm 0.06$; main effect in cueing condition, $F_{1,90} = 25.26$, $p = 2.53 \times 10^{-6}$). Importantly, the cue-induced effect on drift rate varied between sleep and wake groups in a manner consistent with the pattern of ΔWTP: the drift rate was significantly higher for cued relative to uncued items in the sleep but not wake group ((cued vs. uncued) × (sleep vs. wake): $F_{1,90} = 5.29$, $p = 0.024$), suggesting that stimulating the sleeping brain selectively accelerated the drifting process for cued items during the subsequent decision.

Across subjects, the extent to which evidence accumulated faster for cued relative to uncued items can be predicted by the degree of preference bias toward cued items. Specifically, individuals with stronger biases towards cued items in WTP2 demonstrated a more pronounced increment in the drift rate from uncued to cued items ($r = 0.37$, $p = 0.001$; *Figure 4—figure supplement 4A*), with no significant difference in the correlation coefficient across cohorts (sleep: $r = 0.35$; wake: $r = 0.28$; $F_{1,88} = 0.50$, $p = 0.48$). Within sleep group participants, the drift rate estimate was also correlated with brain activity measured during N2 (*Figure 4—figure supplement 4B*). Besides choice behavior, additional posterior predictive simulation was performed to assess whether the between-subject variability in the estimated drift rates was related to the variation observed in RT distributions (*Figure 4—figure supplement 5*).

Whereas sleep exerted no general influence on parameters directly related to valuation, such as drift rate, starting-point variability, and decision threshold (*Figure 4—figure supplement 3*), sleep slowed down the cognitive processing that was unrelated to decision. Relative to the wake group, the sleep group was associated with a longer non-decision time (sleep = $0.72 \pm 0.01$; wake = $0.61 \pm 0.01$; $t = 5.71$, $p = 7.23 \times 10^{-7}$; *Figure 4—figure supplement 3B*), a parameter that captures individual differences in RT affected by cognitive factors separated from valuation (e.g. visual or motor processing). A likely reason for the swift non-decision process in the wake group is that subjects who stayed awake for 90 min felt hungrier than those who took a nap (post-stimulation self-reported hunger in the sleep group = $4.79 \pm 0.29$; wake group = $5.66 \pm 0.26$; $t = -2.26$, $p = 0.026$; *Supplementary file 1*, Table 1A). Consistent with this hypothesis, non-decision time estimates were scaled with the change of self-reported hunger before and after 90-min period at the

across-subject level (r = 0.27, p = 0.009; *Figure 4—figure supplement 4C*), with no significant difference in the correlation coefficient between cohorts (sleep = 0.51; wake = 0.48; F = 0.23, p = 0.633). In contrast, there was no correlation between the non-decision time estimates and the change in vigilance, the familiarity of items, or demographic variables (all p>0.5).

## Discussion

Combining TMR with classic economic paradigms, the present study demonstrates an intriguing connection between sleep and subjective preferences in an important class of decisions widely used for investigating goal-directed behavior. In particular, the study provides behavioral, neural, and modeling evidence indicating that covert cueing during midday nap can alter preferences and choices in a selective manner, independent of more general effects on decision-making such as hunger, vigilance, or choice randomness. The fact that the stimulation effect is sleep-dependent and that cueing in the waking state produces no significant changes in preferences or choices is consistent with past evidence that selective memory reactivation during wakefulness does not always improve memory (*Rudoy et al., 2009*; *Antony et al., 2012*; *Schreiner and Rasch, 2015*) and sometimes even produces opposing consequences (*Diekelmann et al., 2011*). Importantly, and consistent with previous studies (*Antony et al., 2012*; *Oudiette et al., 2013*; *Schreiner and Rasch, 2015*), the degree of post-sleep preference enhancement can be predicted by cue-induced increases of theta or delta power, which have been previously associated with successful cueing during sleep (*Schreiner et al., 2015*; *Schreiner and Rasch, 2015*) and with improved memory consolidation (*Oudiette et al., 2013*; *Batterink et al., 2016*), respectively.

Results of the current study provide novel insights into the nature of cognitive processes that support value evaluation. Our findings corroborate a wealth of evidence suggesting that subjective preferences interact dynamically with the environmental and cognitive states of a decision-maker, and extend these previous findings by demonstrating that sleep also contributes to the dynamic nature of preferences in a flexible, selective manner, such that preferences for specific options can be individually biased during sleep.

In particular, our results are consistent with two broad accounts previously proposed for how preferences could be dynamically modulated. The first involves the possibility that preferences are informed by memory, such that weighting a choice option entails retrieving relevant information from memory at the time of decision-making (*Pennartz et al., 2011*; *Bornstein et al., 2017*). Under this interpretation, preference reflects the outcome of memory retrieval, and verbal cueing during sleep encourages the preference of a familiar, valued item by reactivating and strengthening the relevant memories. For example, sleep TMR may trigger the reactivation of reward-related information associated with the cued snack or assist the consolidation of past episodes, such as purchasing or consuming the cued item. In keeping with this possibility, previous studies have suggested that important memories, including those of motivational salience, are preferentially reactivated during sleep (*Lansink et al., 2008*; *Dunsmoor et al., 2015*; *Blume et al., 2017*) and that successful retrieval of a relevant episodic memory (e.g. a past action (*Bornstein et al., 2017*) or contextual information associated with an action (*Bornstein and Norman, 2017*)) leads to subsequent decision biases. Neurally, this possibility is consistent with evidence supporting the interplay between memory and choice processes, which involves a network of brain regions, including the hippocampal region, in the processing of memory-related value signals (*Barron et al., 2013*; *Gluth et al., 2015*; *Palombo et al., 2015*; *Weilbächer and Gluth, 2016*).

At the computational level, this interpretation is consistent with the proposal that connects preference-guided choices and mnemonic processes under the framework of sequential sampling (*Gluth et al., 2015*; *Shadlen and Shohamy, 2016*). Interestingly, using LBA, a variant of sequential sampling model, we found that sleep TMR selectively accelerated the accumulation of evidence supporting cued items as measured by drift rate. Past theoretical and empirical work has suggested that the drift rate in sequential sampling models reflects the quality of information drawn from stimuli during the evidence accumulation (*Ratcliff and McKoon, 2008*; *Ratcliff et al., 2016*). Within the domain of memory, a higher drift rate is associated with recognizing an item that has been previously presented multiple times relative to an item presented only once (*Ratcliff and McKoon, 2008*; *Gluth et al., 2015*). The correlation between drift rate biases and WTP biases observed in our data thus indicates the intriguing possibility that memory that is selectively reactivated during sleep

renders higher quality evidence signals in favor of the cued item, giving rise to behavioral shift toward items previously named during sleep. Future work employing various process models (*Forstmann et al., 2016*) will be needed to further connect behavioral shifts with measures of memory change before and after TMR to investigate which memory is reactivated (e.g. episodic, semantic, emotional) and how different types of memories differentially affect the choice process.

A second but not mutually exclusive possibility is that rather than playing a direct role in affecting value computation and comparison, verbal cueing initiates a chain of events that exerts modulatory effects on the decision process. For example, the behavioral shift observed in our data may reflect the modulatory effect of memory on attention, which in turn biases preferences and choices at the time of decision-making. In line with this hypothesis, recent evidence based on either eye fixation (*Krajbich et al., 2010*) or cue-approach training (*Schonberg et al., 2014*; *Bakkour et al., 2017*) has suggested the influence of attention on decisions. Consequently, if individuals pay more attention to better-remembered options, preference enhancement following selective memory reactivation during sleep may reflect an increase in attention due to improved memory consolidation (*Chun and Turk-Browne, 2007*; *Goldfarb et al., 2016*; *Rosen et al., 2016*). Under this hypothesis, the drift rate identified in the computational model may be associated with attention bias between cued and uncued items, and possibly scaled with the degree to which attention is affected by memory after verbal cueing. Future work incorporating behavioral probes for changes in attention before and after verbal cueing is needed to test and discriminate between these possibilities. Additional experiments may also include neuroimaging studies to identify neural correlates and functional coupling in brain regions previously implicated in attention, memory, and reward processing.

The results of the present study suggest that sleep likely represents a unique period during which preferences and choices that are otherwise stable can be selectively modified by external cues. This finding is consistent with past evidence that targeted memory reactivation not only benefits memories of newly acquired information but can also be used to eliminate long-standing social biases (*Hu et al., 2015*). It remains to be elucidated whether the findings of the present study are specific to N2 or are generalizable to other stages of sleep, in particular, slow-wave sleep, the primary focus of previous TMR studies. On the one hand, there is evidence for the capacity of neurocognitive processing of verbal stimuli during N2, including extracting information from incoming words (*Kouider et al., 2014*), differentiating their motivational saliency (*Blume et al., 2017*), and reactivating the existing motor or vocabulary memories associated with the words (*Kouider et al., 2014*; *Schreiner and Rasch, 2015*). Accordingly, one might expect that N2 may be particularly suitable for implementing TMR using verbal stimuli, similar to those employed in our study. On the other hand, it is also possible that our results are not restricted to the period of light sleep, given the wealth of evidence on TMR during slow-wave sleep in humans (*Rasch et al., 2007*; *Rudoy et al., 2009*; *Antony et al., 2012*) and the entire non-rapid eye movement sleep in rodents (*Rolls et al., 2013*; *Barnes and Wilson, 2014*). Future studies examining night sleep as opposed to daytime naps or involving deprivation of particular sleep stages are needed to establish the specificity of the observed effects for different sleep stages.

Our results also raise intriguing questions regarding whether the preference enhancement is specific to cued memory reactivation, or hold more generally for both cued and spontaneous reactivation during sleep. According to the active system consolidation model, sleep-dependent memory consolidation critically relies upon repeated reactivation of memory representations (*Diekelmann and Born, 2010*; *Rasch and Born, 2013*). It is therefore possible that the amount of memory reactivation taking place during sleep modulates preference improvement. TMR likely plays a role of triggering and enhancing such reactivation, thereby magnifying the influence on preference in the sleeping brain. Under this hypothesis, memory reactivation, either cued or spontaneous, contributes to the fine-tuning of reward evaluation. Yet unlike cued reactivation, which is predicted to bias the weighting of specific choice option likely through its selective influence on memory traces, spontaneous reactivation may exert a less discriminative impact on valuation. For example, there is evidence that, following overnight sleep, subjects demonstrate an overall increase in favorable perception of the whole choice set, due to enhanced recalls for positive information associated with choice options (*Karmarkar et al., 2017*). In the present study, however, midday nap produces no general effect on either the overt behavior or computational parameters reflecting the latent cognitive processing directly related to decision. One possibility is that the beneficial effect of sleep on memory consolidation requires sufficient sleep duration and the presence of slow-wave sleep

(*Backhaus and Junghanns, 2006*; *Diekelmann et al., 2011*; *Diekelmann et al., 2012*), whereas day-time naps usually consist of lighter sleep stages such as stage 1 and 2 (*Backhaus and Junghanns, 2006*; *Tucker et al., 2006*; *Genzel et al., 2014*). TMR during nap in our experiment may serve to elicit memory reactivation that typically occurs during a longer sleep period. Consistent with this interpretation, previous research shows similar effects of memory stabilization following either 40-min sleep with TMR or 90-min sleep without external stimulation (*Diekelmann et al., 2012*). Future investigations using night sleep or animal models will be needed to assess whether and how sponta-neous memory reactivation affects the storage of reward-related information and to determine whether sleep plays a general role in modulating goal-directed behavior through tuning reward evaluation.

More broadly, simple choices are fundamental for goal-directed behavior and have served as a cornerstone for studying reward and motivational control across species (*Fehr and Rangel, 2011*; *Padoa-Schioppa, 2011*). Results of this study provide novel insights into the link between sleep and preferences elicited by simple choices, thereby pointing to the possibility of shaping a range of behavior through nudging human sleep. Our results, together with previous findings on sleep-dependent reward-related neural activation (*Pennartz et al., 2004*; *Lansink et al., 2008*; *Lansink et al., 2009*; *Perogamvros and Schwartz, 2012b*; *Igloi et al., 2015*), raise the intriguing question regarding whether, and under what circumstances, value assessment and comparison can be perturbed by external cues while asleep. Future studies are needed to address whether our find-ings generalize to other types of preferences, including those involving risk and ambiguity (*Hsu et al., 2005*), temporal discounting (*McClure et al., 2004*), loss aversion (*Tom et al., 2007*), and prosocial considerations (*Hein et al., 2016*).

## Materials and methods

### Subjects

Ninety-nine subjects were recruited by the Laboratory of Sleep Research at the Institute of Mental Health, Peking University (79 women, age = 23.42 ± 2.36; *Supplementary file 1*, Table 1A). All sub-jects provided informed consent approved by the Ethics Committee at Peking University. All sub-jects were native Chinese speakers and had normal or corrected-to-normal vision and hearing. They were not on a diet or taking any medications that would interfere with the experiment, and they had no history of neurological or psychiatric illnesses, including any sleep-related or eating-related disor-ders. All subjects habitually napped in the afternoon and followed a normal sleep-wake rhythm (i.e. no shift work for at least 1 week before the experiments and at least 6 hr sleep per night on average).

Seven subjects were excluded from the analyses due to insufficient variances in WTP1 (N = 3), a priori–defined insufficient nap time (N = 2), or being able to recall verbal stimuli in the post-sleep awareness test (N = 2). Subjects were randomly divided into sleep (N = 47) and wake (N = 45) groups. We recorded sleep using polysomnography in all sleep group participants, and cue-induced EEG signals in a subset of 23 sleep group participants (*Supplementary file 1*, Table 1D, Table 1E).

### Procedure

All subjects were instructed to refrain from eating for 4 hr before coming to the laboratory and were provided with identical amounts of food upon arrival to ensure a minimal level of hunger. Snack items used in the study were selected based on independent ratings of familiarity, level of valence, and arousal collected in a pilot study with another 49 subjects (*Supplementary file 1*, Table 1B). Subjects were not informed about the study hypotheses and did not know that they were going to receive any external stimulation while asleep.

The experiment started at noon. Subjects were instructed that one decision would be randomly selected from each of the three tasks and implemented at the end of the experiment and that if a snack were purchased in the selected decision, the participant would be required to consume the snack in the lab. Subjects were provided with ¥15 that could be used for purchase at the beginning of each task and were paid an additional ¥250 upon completion of the study. After receiving instruc-tions, all participants indicated WTP for 60 snack items presented in a random order in a BDM auc-tion. Subjects were then randomly divided into sleep and wake groups and received verbal

stimulation. In the sleep group, verbal cueing started 2 min after a subject displayed stable N2 sleep for the first time and paused if there was any polysomnographic signal indicating microarousal, awakening, or entering other sleep stages. Wake group participants were kept awake for the same duration and were presented with verbal stimuli at approximately the same time when cues were delivered to sleeping participants. During breaks before and after the stimulation session, we collected self-reports of vigilance and hunger levels from all subjects. After that, all participants performed the same BDM auction again, followed by binary choices between pairs of cued and uncued items. Finally, subjects finished a number of post-experiment questionnaires aiming at evaluating the awareness of cue delivery during sleep and familiarity of snack items. All experimental stimuli and behavioral data acquisition were performed using MATLAB (MathWorks) with Psychotoolbox (*Brainard, 1997*).

For the wake group, participants were allowed to engage in quiet activity including reading books or newspapers, watching documentary films, etc. during the 90-min stimulation period. An experimenter accompanied these participants to ensure that they stayed awake throughout the period. When verbal cues were delivered, wake group participants were instructed to pause their activity and pay full attention to the stimuli. No other task was performed during this period and no EEG was recorded on these participants.

## Auditory stimuli

We hired a professional male broadcaster to record all verbal stimuli. Sounds were recorded, edited, and volume-level smoothed using Audacity (http://audacity.sourceforge.net/). Verbal stimuli (duration range: 0.65–1.05 s, mean ± SD: 0.89 ± 0.08) were delivered via loudspeaker during auctions (intensity: 70 dB) and the verbal stimulation sessions (intensity: 55 dB). A low-intensity, 35 dB white noise was constantly delivered during sleep to block environmental noise.

After sleep group subjects entered N2 as assessed by standard EEG criteria (*Iber et al., 2007*), verbal cues were delivered with intervals ranging from 4.5 s to 5.0 s. To increase the chance of triggering memory replay in the sleep group, each snack name was repeated for exactly 10 times in a row before presenting the next snack item. For each subject, eight different snack names were presented in a randomized order. Verbal stimulation was immediately paused when EEG recordings showed any signs of micro-arousal, awakening, or changing of sleep stages. On average, cueing was interrupted 0.75 ± 0.58 (mean ±SD) times in the sleep group.

## EEG recordings

Six scalp electrodes (C3, C4, F3, F4, O1, and O2) were placed according to the international 10–20 system and referenced to the contralateral mastoid, along with electrooculogram (EOG) and chin electromyogram (EMG) channels. Impedances were kept below 5 kΩ for EEG electrodes and below 10 kΩ for EOG and EMG electrodes. EEG signals were sampled online at 500 Hz with Profusion Net Beacon software (Compumedics Sleep Study System, Melbourne, Australia) and filtered between 0.3 and 35 Hz. Thirty second epochs were used for manual analysis, and periods of wakefulness, N1, N2, N3, and REM sleep were identified offline by two independent raters based on standard criteria from the American Academy of Sleep Medicine (AASM) (*Iber et al., 2007*).

## EEG data analyses

EEG data were preprocessed and analyzed using customized MATLAB scripts (*Yin and Ai, 2018*) and EEGLAB toolbox (*Delorme and Makeig, 2004*). These data segments were visually identified and excluded if they contained arousal, motor, or other artifacts. To analyze event-related potentials (ERPs), we segmented EEG data into 2000 ms epochs starting from 200 ms before the stimulus onset. The interval from −200 to 0 ms with respect to the stimulus onset was used for baseline correction. EEG signals were averaged across all verbal stimuli and all subjects for each electrode. This procedure yielded a K-complex-like (KCs) waveform at the electrodes of F3, F4, C3 and C4, but not for the electrodes of O1 and O2 (*Figure 4—figure supplement 1A*). We calculated the averaged amplitude of the early component (positive, KC+) of KCs in a time window between 200 and 600 ms and the later component (negative, KC-) between 600 and 1200 ms (*Blume et al., 2017*). Repeated measures ANOVA showed no significant difference among these four electrodes in amplitudes of

the KCs (*Supplementary file 1*, Table 1F), thus we presented the ERP of the electrode F3 for illustrating EEG-related results.

To examine event-related changes in spectral power (ERSPs), raw EEG data were segmented into artifact-free epochs ranging from −1000 to 2000 ms relative to the stimulus onset. Power spectral estimation of the EEG signal was achieved using the fast Fourier transform with Hanning window tapering with a window size of 400 ms (*Rasch et al., 2007*; *Rudoy et al., 2009*; *Hauner et al., 2013*). To avoid edge effects, trials were segmented from −800 to 1800 ms with respect to the stimulus presentation. This was accomplished by discarding 200 ms at the beginning and the end of trials (*Schreiner and Rasch, 2015*; *Blume et al., 2017*). Frequency bands were defined as follows: delta (0.5–4 Hz), theta (4–8 Hz), alpha (8–12 Hz), beta (12–30 Hz), and gamma (30–35 Hz). To assess the effect of verbal stimuli on sleep depth, we compared the mean spectral power following the stimulus onset with the mean power of the pre-stimulus time window at each frequency using two-tailed, paired t-tests (*Figure 4—figure supplement 1B*) (*Rudoy et al., 2009*; *Hauner et al., 2013*; *Ruch et al., 2014*). The subject-level cue-induced power was computed as the mean spectral power by averaging frequency spectra of all items within each subject. Similarly, the item-level cue-induced power was computed as the mean spectral power by averaging frequency spectra of all sleep group subjects given a cued item. We then subtracted the corresponding frequency spectra power respect to that during the prestimulus time window from - 800 to - 200 ms, based on which we performed correlation analyses between cue-induced power change and preference changes at both the individuals and items.

## Post-experiment questionnaire

At the end of the experiment, we administered a number of survey questions evaluating (i) the awareness of cue delivery during sleep and (ii) familiarity of snack items. The post-sleep awareness test was adopted from previous studies (*Rudoy et al., 2009*; *Antony et al., 2012*; *Ai et al., 2015*) using similar TMR procedures ('Did you hear anything during sleep? Answer: Yes (please clarify)/ No'), based on which two sleep group participants who reported having heard sounds were removed from data analyses. Besides the awareness test which was administered within the sleep cohort, the post-experiment survey also included questions for all participants assessing how familiar subjects were with each snack item and whether or not subjects had previously consumed each item. Results of these questions were largely consistent with findings of the pilot study based on which snacks were selected and were included in statistical analyses testing the robustness of our findings (*Supplementary file 1*, Table 1B, C).

## Computational modeling

We used the LBA, a variant of race model that provides a simple characterization of noise in the drift process and a closed-form likelihood function for choices and RT that is computationally efficient (*Brown and Heathcote, 2008*). The model has been previously applied to a range of binary or multiple-choice decisions, including value-based (*Trueblood et al., 2014*; *Rodriguez et al., 2015*) and memory-informed choices (*Hawkins et al., 2016*). Following previous studies (*Brown and Heathcote, 2008*; *Forstmann et al., 2008*; *Forstmann et al., 2010*), our LBA model represents a choice between cued and uncued snack items as a race between two independent signals, each for one snack item, that linearly and deterministically accumulate evidence until one accumulator reaches a threshold.

Specifically, for each decision, accumulators for cued and uncued items start from their own initial point $a_i$ (where $i$ = cued or uncued), where each $a_i$ is sampled independently from an identical uniform distribution $U(0,A)$ and $A$ represents the between subject variability in the starting point. Accumulator $i$ increases over time at the drift rate $v_i$ drawn from a corresponding normal distribution $N(k_i, s)$, where parameter $k_i$ indicates the mean of the normal distribution for accumulator $i$ and parameter $s$ represents the standard deviation common for accumulators of both cued and uncued items. The higher the drift rate $v_i$ the faster the evidence can be accumulated in support of option $i$. A decision favoring one particular item is made at time $T$ when the accumulator for this item exceeds the decision threshold $b$. Reaction time is defined as $RT = T + NDT$, where parameter $NDT$ represents non-decision time.

To calibrate the model with observed data, we used a hierarchical-Bayesian model estimation method (*Kruschke, 2010*), assuming that individual parameters were randomly drawn from distributions governed by a set of group-level parameters that were sampled independently from the corresponding prior distributions (*Lee and Wagenmakers, 2014*) (*Supplementary file 1*, Table 1H). We computed the posterior likelihood of observing choice and reaction time data with the Markov chain Monte Carlo (MCMC) method implemented in RStan (*Carpenter et al., 2017*). Three MCMC chains were simulated with 20,000 iterations after 20,000 burn-ins, resulting in 20,000 posterior samples for each parameter. All estimated parameters were checked for convergence both visually (from the trace plot) and through the Gelman–Rubin test (All $\widehat{R} < 1.1$) (*Gelman et al., 2014*).

## Data and code availability

The data and code reported in this paper have been deposited in Open Science Framework (https://osf.io/9ndhy/).

## Acknowledgements

This work was supported by National Natural Science Foundation of China (31571099, U1402226, 81801315, 31671171, and 31630034), Beijing Municipal Science & Technology Commission (Z161100002616006 and Z181100001518005), and the National Basic Research Program of China (2015CB856404 and 2015CB553503). The authors thank Xijian Dai and Dr. Wangshu Feng for assistance in data collection and Drs. Larry D. Sanford, Jian Li, Xiaohong Wan, Xiaolin Zhou, Bo Shen, and Jie Hu for constructive comments.

## Additional information

### Funding

| Funder | Grant reference number | Author |
| --- | --- | --- |
| National Basic Research Program of China | 2015CB856400 | Jie Shi |
| National Basic Research Program of China | 2015CB553503 | Jie Shi |
| National Natural Science Foundation of China | 31571099 | Jie Shi |
| National Natural Science Foundation of China | U1402226 | Jie Shi |
| Beijing Municipal Science and Technology Commission | Z181100001518005 | Jie Shi |
| Beijing Municipal Science and Technology Commission | Z161100002616006 | Jie Shi |
| National Natural Science Foundation of China | 81801315 | Sizhi Ai |
| National Natural Science Foundation of China | 31671171 | Lusha Zhu |
| National Natural Science Foundation of China | 31630034 | Lusha Zhu |

The funders had no role in study design, data collection, and interpretation, or the decision to submit the work for publication.

### Author contributions

Sizhi Ai, Conceptualization, Data curation, Software, Formal analysis, Investigation, Visualization, Writing—original draft, Writing—review and editing; Yunlu Yin, Conceptualization, Software, Formal analysis, Investigation, Visualization, Methodology, Writing—original draft, Writing—review and editing; Yu Chen, Data curation, Investigation; Cong Wang, Software, Formal analysis; Yan Sun,

Conceptualization, Methodology; Xiangdong Tang, Validation, Writing—review and editing; Lin Lu, Resources, Validation, Methodology; Lusha Zhu, Conceptualization, Formal analysis, Supervision, Validation, Visualization, Methodology, Writing—original draft, Writing—review and editing; Jie Shi, Conceptualization, Resources, Supervision, Funding acquisition, Validation, Methodology, Project administration, Writing—review and editing

### Author ORCIDs
Yunlu Yin ![iD] http://orcid.org/0000-0002-6433-3870
Cong Wang ![iD] https://orcid.org/0000-0001-7997-4200
Lusha Zhu ![iD] https://orcid.org/0000-0001-8717-6356
Jie Shi ![iD] http://orcid.org/0000-0001-6567-8160

### Ethics
Human subjects: All participants provided written informed consent. Study procedures were reviewed and approved by the Ethics Committee at Peking University.

### Decision letter and Author response
Decision letter https://doi.org/10.7554/eLife.40583.018
Author response https://doi.org/10.7554/eLife.40583.019

## Additional files
### Supplementary files
• Supplementary file 1. Supplementary Tables 1A through 1I (A) Table 1A. Subject information. All participants were asked to report their subjective level of hunger (from 1 = 'not hungry at all' to 8 = 'very hungry') and vigilance (from 1 = 'very vigilant, not sleepy at all' to 7 = 'very sleepy, taking great efforts to keep awake') before and after the verbal stimulation. Parentheses contain standard deviations. p-Values are calculated using two-tailed Student T test for comparing the difference in mean of each variable between the sleep and wake groups. Parentheses contain standard deviations. (B) Table 1B. Snack items included in the study together with their English translations. Snack items were selected based on a pilot experiment in which 49 subjects were recruited to assess the familiarity, valence, and subjective arousal (Self-Assessment-Manikin scale) of a pool of candidate snacks. Based on those ratings, we selected 60 items with median familiarity (mean ±SD: 3.51 ± 0.84), positive valence (mean ±SD: 5.08 ± 1.26), and median arousal level (mean ±SD: 4.65 ± 1.33) (associated with *Figure 1*). (C) Table 1C. Linear regressions on the effect of verbal cueing for sleep and wake groups after controlling for differences in age, gender, BMI, as well as self-reported familiarity and differences in vigilance and hunger before and after the cueing session. The dependent variable in the first regression is equal to the average difference of ΔWTP between cued and uncued items for each subject. The dependent variable in the second regression is equal to the likelihood of choosing cued over uncued item in the binary decision task for each subject. *p < 0.05; **p < 0.01; ***p < 0.001, two-tailed (associated with *Figure 2*). (D) Table 1D. Durations of sleep stages (in minutes) for sleep group subjects with or without ERP. There is no significant difference in the duration of sleep stages. Parentheses contain standard errors. p-Values were calculated using two-tailed Student T test for between-group comparisons (associated with *Figure 4*). (E) Table 1E. Effects of cueing on preferences and choices in sleep group subjects with or without ERP. There is no significant difference in the effect of verbal stimulation on either the ΔWTP or choice behavior in the binary decision task. Parentheses contain standard errors. p-Values are calculated using two-tailed Student T test for between-group comparisons (associated with *Figure 4*). (F) Table 1F. Average EEG amplitudes of positive/negative components of the K-complex-like evoked responses (KCs) at each electrode. We averaged the EEG amplitudes measured over the interval from 200 to 600 ms (for KC+) and from 600 to 1200 ms (for KC-) for each subject at each electrode. We found significant KC+/KC- at the frontal electrodes F3 and F4, and the central electrodes C3 and C4 (Student T-test, N = 23, all p values are Bonferroni corrected). For electrodes with significant KCs (F3, F4, C3, and C4), repeated measures ANOVA showed no significant difference among these four electrodes in both KC+ ($F_{3,84}$ = 0.528, p = 0.664) and in KC- ($F_{3,84}$ = 0.528, p = 0.664). *p < 0.05; **p < 0.01; ***p < 0.001, two-

tailed (associated with *Figure 4A*). (G) Table 1G. Pearson correlations between ΔWTP and averaged cue-induced power for each frequency band at either subject- or item-levels. For across subject analysis, we examined the correlation between the ΔWTP and the power change of cue-induced band that were averaged over all cued items within each subject. For across item analysis, we examined the correlation between the ΔWTP and the power change of cue-induced band that were averaged over all subjects given the same cued item. All p-values are Bonferroni corrected. *p < 0.05; **p < 0.01; ***p < 0.001, two-tailed (associated with *Figure 4B–D*). (H) Table 1H. Priors used in the hierarchical Bayesian model estimation for LBA. We performed model estimation under the assumption that individual parameters are drawn from distributions for the sleep and wake group separately, with group level parameters sampled from joint prior distributions (associated with *Figure 4—figure supplements 1–4*). (I) Table 1I. Percentiles of RT distributions in panel B-C. These data show that a high drift rate bias (therefore high reactivation during N2) is associated with a more prominent change in the tail of the RT distribution and a smaller change in the leading ledge (associated with *Figure 4—figure supplements 1–4*).
DOI: https://doi.org/10.7554/eLife.40583.013

• Transparent reporting form
DOI: https://doi.org/10.7554/eLife.40583.014

## Data availability

Data and code used for data analysis are publicly available online via Open Science Framework (OSF) at (https://osf.io/9ndhy/).

The following dataset was generated:

| Author(s) | Year | Dataset title | Dataset URL | Database and Identifier |
| --- | --- | --- | --- | --- |
| Yin Y, Ai S | 2018 | Data and Code for Promoting subjective preferences in simple economic choices during nap | https://osf.io/9ndhy/ | Open Science Framework, 10.17605/OSF.IO/9NDHY |

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
