## [Decision Letter]

Thank you for submitting your article "Promoting subjective preferences in simple economic choices during sleep" for consideration by *eLife*. Your article has been reviewed by two peer reviewers, including Michael Breakspear as the Reviewing Editors and Reviewer #1, and the evaluation has been overseen by a Reviewing Editor and Michael Frank as the Senior Editor. The following individual involved in review of your submission has agreed to reveal their identity: Björn Rasch (Reviewer #2).

The reviewers have discussed the reviews with one another and the Reviewing Editor has drafted this decision to help you prepare a revised submission.

Summary:

This is an interesting, well-posed and clearly reported study. The finding that cues provided during sleep influence behaviour according to their neurophysiological correlates is likely to be of substantial interest to sleep researchers and those examining value-based decision making.

The authors examined the effect of target memory reactivation (TMR) during a 90 min midday nap on choice preferences in economic decision paradigms. They show that re-exposure to names of snacks during NREM sleep biased choices towards the snack. In contrast, re-exposure to the snack names during wakefulness did not affect decision-making. In addition, stimulus-evoked δ activity and theta activity predicted changes in choice behavior on the both the subject and item level. The authors conclude that choice preferences might relate to memory processes, which can be biased using TMR techniques during sleep.

The paper and the reported findings are exciting. While the memory benefits of TMR during sleep are well established, the authors show for the first time that also behavior in decision-making paradigms can be biased by influencing memory processing during sleep. The authors use established paradigms and cueing techniques in their study, and the methodological approach is sound. The participant samples are relatively large compared to other TMR studies. A particular strength of the paper is that the authors show that the effect of TMR on decision-making occurs in the sleep, but not in the wake group. Furthermore, the authors provide a wide range of analyses and results, which are all highly informative and important.

Essential revisions:

Both reviewers' concerns are listed separately but note that comment 3 from reviewer 1 is very similar to comment 2 from reviewer 2 and comment 2 by reviewer 1is similar to comment 4 by reviewer 2: it is reasonable for you to collapse your responses to these concerns into two single responses.

Reviewer #1:

1) Were the subjects told they were going to be delivered verbal cues when asleep? Were they aware of the study hypotheses? What was the nature and results of the "post-sleep awareness test" – perhaps move this to the Results section.

2) What was the nature of recruiting so many participants whom regularly napped in the afternoon and duly cooperated and slept in a laboratory setting? This does not seem a typical population to me. How were the non-sleep group kept awake and were they checked for drowsiness during the word cueing?

3) What motivated the decision to multiply N2 by the mean δ and theta amplitudes? This sounds rather exploratory to me (given that each measure alone is close to significant; that several various combinations are possible; and that the final result (p=0.026) would not survive multiple comparisons.

Reviewer #2:

1) As the general results of cueing during sleep appear to be rather robust, it puzzles me that the authors did not find any differences in the drift diffusion model. Is it possible to compare cued vs uncued items in this model, and not sleep vs. wake in general? How do the authors explain that this model is not sensitive for the observed choice preferences? Please discuss this point in more detail in the discussion. In particular, please elaborate on the functional consequences of the null findings in this model.

2) Related to the point above, the reported correlation in Figure 4F (i.e., N2 amount*delta activity and drift rate) appears to be highly exploratory and speculative. With four model parameters and many possible combinations of sleep stage durations and oscillatory activity, it is highly probable that this is a false positive. Did the authors also correct for multiple comparisons here, as in the other analyses? I would strongly suggest omitting this part or at least strongly attenuating the conclusions made from this analysis. For me, the more important point here is that the model parameters are not sensitive for cueing benefits.

3) The authors should give some more details on the reactivation itself: How many cues were delivered on average? How many cues were delivered in N2 vs N3 vs in other sleep stages? How does number of cues relate to changes in choice preferences? Was the same snack name presented 10 times in a row or intermixed with other snack names? Please specify.

4) Also, please give more details on the wake control group. What did this group do while listening to the stimuli? Did they perform on a different task or did they activity attend the presented stimuli? Was EEG recorded during that time? If yes, please report also the results.

5) In general, the authors also need to make clear that they did not find any general effect of sleep on choice preferences in their study. Mechanistically, this is an important point. In the sleep and memory field, it is assumed that memories are spontaneously reactivated during sleep, and that TMR enhances this spontaneous mechanism. If choice preference depends on the degree of reactivation during sleep, then the change in WTP should be generally stronger after sleep as compared to wakefulness. (However, some results also in the sleep/memory domain suggest that general sleep associated benefits for memory are only visible after longer sleep periods, whereas TMR benefits are detectable already after short naps, see Diekelmann et al., 2012,). Thus, please discuss what general effects of sleep / spontaneous reactivations during sleep you would expect on choice preference and why. In other words, does memory consolidation during sleep generally affect decisions / preferences, or is TMR simply a way of (artificially) biasing a process that does not depend on sleep itself.

---

## [Author Response]

Essential revisions:Both reviewers' concerns are listed separately but note that reviewer 1 comment 3 is very similar to comment 2 from reviewer 2 and reviewer 1 comment 2 is similar to comment 4 by reviewer 2: it is reasonable for you to collapse your responses to these concerns into two single responses.Reviewer #1:1) Were the subjects told they were going to be delivered verbal cues when asleep? Were they aware of the study hypotheses? What was the nature and results of the "post-sleep awareness test" – perhaps move this to the Results section.

We thank the reviewer for urging us to clarify this information. Critical to our experimental design, subjects were not informed about the study hypotheses and did not know that they were going to receive any external stimulation while asleep. Instead, the experiment was presented as a study aiming at measuring snack preferences with or without napping.

Second, at the end of the experiment, we administered a number of survey questions evaluating (i) the awareness of cue delivery during sleep and (ii) familiarity of snack items. The post-sleep awareness test was adopted from previous studies using similar TMR procedures (“Did you hear anything during sleep? Answer: Yes (please clarify)/No”. Rudoy et al., 2009; Antony et al., 2012; Ai et al., 2015), based on which 2 sleep group participants who reported having heard sounds were removed from data analyses.

Besides the awareness test, which was administered within the sleep cohort, the post-experiment survey also included questions for all participants assessing how familiar subjects were with each snack item and whether or not subjects had previously consumed each item. Results of these questions were largely consistent with findings of the pilot study based on which snacks were selected, and were included in statistical analyses testing the robustness of our findings (Supplementary file 1, Table 1B, Table 1C).

We have now included this additional information in Materials and methodssection.

2) What was the nature of recruiting so many participants whom regularly napped in the afternoon and duly cooperated and slept in a laboratory setting? This does not seem a typical population to me. How were the non-sleep group kept awake and were they checked for drowsiness during the word cueing?

Per editor suggestion, we combined question 2 from reviewer 1 and question 4 from reviewer 2, which are both about subjects in our study but from two distinct aspects: habitual nappers, and the experimental procedure for wake group subjects. We address each point in turn.

On the former, our choice of habitual nappers for the study was informed by (i) previous TMR studies, (ii) availability of these subjects, and (iii) experimental procedure specifically designed for these subjects.

First, our study, with subjects who regularly nap in the afternoon, is in keeping with previous TMR studies focusing on daytime naps. Creery et al. (2015), for example, recruited subjects “who anticipated that they would be able to sleep in the afternoon” and Ai et al., 2015 used subjects who “slept habitually for 1/2–2 h in the afternoon”. Prior studies with no restriction on napping habits often involve removing a significant number of participants due to the difficulty of these participants in falling asleep during the sleep phase of the experiment (e.g., Antony et al., 2012). To overcome this difficulty, previous research has also requested participants to sleep a few hours less than normal before coming to the experiment (Hauner et al., 2013). However, partial sleep restriction is likely to affect eating behavior on the subsequent day, including increasing caloric intake and self-reported hunger (Brondel et al., 2010). Importantly, such an effect is likely to interact with intrinsic food preferences and the caloric density of food options (Greer et al., 2013), therefore seriously interfering with the purpose of our study.

Another important reason for choosing habitual nappers in the current study is that daytime napping is not uncommon in China, especially among college students. According to a large-scale, cross-country survey on sleeping habits around the world (Soldatos et al., 2005), 33% of Chinese general population regularly naps in the afternoon, a percentage that is significantly higher than the average number around the world. Relative to the general population, it has been observed that college students are more likely to take a midday nap for preventing reduced vigilance in the afternoon (Milner and Cote, 2009), an impression that is supported by a study suggesting that 53% of undergraduate students in Australia reported frequent napping (Lovato et al., 2014).

In addition, as reviewer 1 pointed out, when focusing on habitual nappers in the experiment, it is of critical importance to make sure that our wake group participants do not fall asleep over the course of the stimulation session. During this 90-min period, the wake group subjects were allowed to engage in quiet activity including reading books or newspapers, watching documentary films, etc. Importantly, an experimenter accompanied these participants to ensure that they stayed awake throughout the period. In addition, using the Stanford Sleepiness Scale (Diekelmann et al., 2011), we assessed self-reported alertness for sleep and wake cohorts, and found no significant differences in vigilance after the 90-min period between two groups (sleep = 2.21 ± 0.16; wake = 2.76 ± 0.25; t = -1.84, P = 0.07; using a 7-point scale with 1 being very vigilant and 7 being very sleepy; Supplementary file 1, Table 1A, Table 1C).

We also thank reviewer 2 for urging us to clarify the experimental procedure related to the wake group. During the 90-min period, the wake group participants were allowed to engage in quiet activity including reading books or newspapers, watching documentary films, etc. When verbal cues were delivered, wake group participants were instructed to pause their activity and pay full attention to the stimuli. No other task was performed and no EEG was recorded during this period. We have now included this additional information in the Materials and methods

References:

Creery JD, Oudiette D, Antony JW, Paller KA (2015) Targeted memory reactivation during sleep depends on prior learning. Sleep 38:755-763.

Brondel L, Romer MA, Nougues PM, Touyarou P, Davenne D (2010) Acute partial sleep deprivation increases food intake in healthy men. Am J Clin Nutr 91:1550-1559.

Milner CE, Cote KA (2009) Benefits of napping in healthy adults: impact of nap length, time of day, age, and experience with napping. J Sleep Res 18:272-281.

Lovato N, Lack L, Wright H (2014) The napping behaviour of Australian university students. PLoS One 9:e113666.

3) What motivated the decision to multiply N2 by the mean δ and theta amplitudes? This sounds rather exploratory to me (given that each measure alone is close to significant; that several various combinations are possible; and that the final result (p=0.026) would not survive multiple comparisons.

Per editor suggestion, we combined questions from reviewer 1 and 2 that are both related to multiplying N2 length by δ activity for examining the neurophysiological correlate of the drift rate.

Inspired by Hu et al., 2015, in which the product of SWS and REM duration was used to predict the behavioral effect following TMR, we explored how δ activity and N2 duration could be combined to explain individual differences in post-sleep binary decisions. As the reviewers pointed out, this investigation was explorative in nature and not corrected for multiple corrections. Following reviewers’ suggestions, we have now removed it from the manuscript (also see the reply to point 1 of reviewer 2).

Reviewer #2:1) As the general results of cueing during sleep appear to be rather robust, it puzzles me that the authors did not find any differences in the drift diffusion model. Is it possible to compare cued vs uncued items in this model, and not sleep vs. wake in general? How do the authors explain that this model is not sensitive for the observed choice preferences? Please discuss this point in more detail in the discussion. In particular, please elaborate on the functional consequences of the null findings in this model.

We thank the reviewer for raising this important point and the suggestion for investigating the difference between cued vs. uncued items, rather than sleep vs. wake groups, using established process models of choices. We address this issue by estimating binary decision data with another variant of sequential sampling model, the Linear Ballistic Accumulation (LBA) model. Unlike DDM, this model allows us to (i) directly compare the evidence accumulation processes between cued and uncued items, and (ii) perform a more stringent test on the latent cognitive components giving rise to decisions.

First, as reported in the previous submission, DDM makes specific assumptions about the underlying decision process that only one parameter, the initial bias, would reflect the potential difference between cued and uncued items. Other parameters, such as drift rate, decision threshold, and non-decision time, are assumed to be common or reflect a unified process for alternative options. According to our previous estimation, there was no significant bias either towards or against cued items in the initial drift point for both subject groups (sleep = 0.48 ± 0.04, P = 0.108, t-test for whether initial bias is different from.5, the value of zero bias; wake = 0.49 ± 0.04, P = 0.109, t-test for initial bias different from.5), suggesting symmetric decision criteria (aka response caution) for cued and uncued snacks.

Based on DDM alone, however, we cannot rule out the possibility that TMR affects other aspects of the post-sleep choice process that is not captured by the initial bias, such as the rate of evidence accumulation for cued items. Making such an inference would involve characterizing the drifting process for each available option separately, yet DDM assumes a single accumulator reflecting the relative value for competing options. We therefore used LBA, which, similar to DDM, is a popular sequential sampling model for characterizing choice and RT in a range of behavior including value-based choices (Brown and Heathcote, 2008; Donkin et al., 2011). Unlike DDM, however, LBA assigns separate accumulator for each available option racing towards a common decision threshold. When one accumulator reaches the threshold, the decision corresponding to that accumulator is made. Importantly, accumulators for alternative options are assumed to be independent, each governed by a different drift rate. This allows us to separately identify and directly compare drift rate estimates for items that have vs. have not been previously stimulated.

A second reason for adopting LBA is that a rigorous comparison between cued and uncued items would involve inferring latent cognitive components from overt behavior without using subjective preferences as model inputs. Ideally, preferences or factors related to preferences should be recovered from decision and RT, rather than entered into the model as observables. However, following prior studies on DDM in value-based decisions (e.g., Hutcherson et al., 2015; Polania et al., 2015), WTP2 was entered into DDM for computing trial-level value difference (i.e., *DV = WTP2_cued_ – WTP2_uncued_*). LBA model, on the other hand, offers a simpler characterization of noise and a computationally more efficient likelihood function, which facilitates the parameter estimation based only on the data of choice and RT. This allows us to test whether the value of drift rate revealed solely based on choice and RT in the binary decision task would reproduce patterns of WTP elicited from the separate auction task.

Following prior studies (Brown and Heathcote, 2008), our LBA model included five parameters: two drift rates for cued and uncued items respectively, starting point variability, decision threshold, and non-decision time. Results based on a hierarchical Bayesian model estimation suggested that LBA explained observed patterns in RT and choice behavior very well: the observed choices and RT were highly correlated with predictions based on model estimation, with both correlation coefficients close to 1 (RT: r = 0.995, P = 2.2 × 10^-16^; choices: r = 0.984, P = 2.2 × 10^-16^; now included in Figure 4—figure supplement *2*) and no significant difference in the explanatory power between sleep and wake groups (RT: F_1,88_ = 0.42, P = 0.52; choices: F_1,88_ = 0.50, P = 0.48). Comparing with DDM used in our previous submission, LBA provides better explanation for choice and RT as indicated by either WAIC (DDM = 3768.3, LBA = 3528.9) or Leave-one-out Cross-Validation (DDM = 3854.8, LBA= 3696.9).

Consistent with our hypothesis, drift rate estimates derived from LBA demonstrated a pattern similar to that of ΔWTP (as shown in Figure 2A). There was no main effect in the drift rate between sleep and wake groups (sleep = 3.17 ± 0.06; wake = 3.17 ± 0.07; main effect in treatment, F_1,90_ = 0, P = 1), yet we found a significant within-subject difference between cued and uncued items, such that previously cued items were associated with higher drift rates (cued = 3.37 ± 0.06; uncued = 2.96 ± 0.06; main effect in cueing condition, F_1,90_ = 25.26, P = 2.53×10^-6^). Importantly, the cue-induced effect on drift rate varied between sleep and wake groups in a manner consistent with the pattern observed in ΔWTP: the drift rate was significantly higher for cued relative to uncued items in the sleep but not wake group ((cued vs. uncued) × (sleep vs. wake): F_1,90_ = 5.29, P = 0.024).

Across subjects, the extent to which evidence accumulates faster for cued relative to uncued items can be predicted by the degree of preference bias toward cued items. Specifically, individuals with stronger biases towards cued items in WTP2 demonstrated a more pronounced increment in the drift rate from uncued to cued items (r = 0.37, P = 0.001; now included in Figure 4—figure supplement 4A), with no significant difference in the correlation coefficient across cohorts (sleep: r =.28; wake: r =.35; F = 0.50, P = 0.48). To validate these results, we performed posterior predictive simulation to assess, besides choice behavior, the extent to which drift rate variability was related to the variation in RT distributions (now included in Figure 4—figure supplement 5).

Interestingly, and consistent with results of DDM reported in the initial submission, the drift rate difference between cued and uncued items was significantly correlated with a neurophysiological measure we previously identified –– the product of N2 length by δ activity (r = 0.50, P = 0.015). Following reviewers’ suggestion, we no longer highlight this result in the main text (now included in Figure 4—figure supplement 4B).

Besides drift rate, LBA offers interesting findings on a new aspect of the choice process, which may further help to validate modeling results. Relative to the wake group, the sleep group was associated with a longer non-decision time (sleep = 0.72 ± 0.01; wake = 0.61 ± 0.01; t = 5.71, P = 7.23 × 10^-7^), a parameter that captures individual differences in RT affected by cognitive factors unrelated to valuation (e.g., visual or motor processing). A likely reason for the swift non-decision process in the wake group is that subjects who stayed awake for 90 min felt hungrier than those who took a nap (post-stimulation self-reported hunger in the sleep group: 4.79 ± 0.29; wake group: 5.66 ± 0.26; t = – 2.26, P = 0.026). Consistent with this hypothesis, non-decision time estimates were scaled with the change of self-reported hunger before and after 90-min period at the across-subject level (r = 0.27, P = 0.009; Figure 4—figure supplement 4C), with no significant difference in the correlation coefficient between cohorts (sleep =.51; wake =.32; F = 0.23, P = 0.63). In contrast, there was no correlation between the non-decision time estimates and the change in vigilance, the familiarity of items, or demographic variables (all P > 0.5).

Together, these results support a multi-accumulator sequential sampling model, according to which the accumulation of evidence in support for previously cued options is accelerated by sleep TMR. More broadly, the above modeling results point to a possibility that memory that is reactivated during sleep renders higher quality evidence signals in favor of the associated item during the post-sleep choice process.

In addition, these results suggest that, whereas sleep slows down the cognitive processing unrelated to decisions, sleep likely does not exerts a general influence on parameters directly associated with value computation and comparison in our task, including drift rate, decision threshold, and starting-point variability. Notably, we found no significant main effect in LBA drift rate estimates between the sleep and wake group, paralleling our previous finding that the sleep/wake treatment does not affect DDM drift rates, and that sleep alone has no overall effect on choice preferences as indexed by WTP.

Compared with previous results based on DDM, we feel that the current LBA model provides a better description for the underlying cognitive processing and how such processing is influenced by TMR. We thus replace DDM with LBA in the main text of the revision, with all supporting figures and tables in the supplement (Figure 4—figure supplement 2-5; Supplementary file 1, Table1H).

References:

Donkin C, Brown S, Heathcote A, Wagenmakers EJ (2011) Diffusion versus linear ballistic accumulation: Different models but the same conclusions about psychological processes? Psychon Bull Rev 18:61-69.

Hutcherson, C.A., Bushong, B., Rangel, A., 2015. A Neurocomputational Model of Altruistic Choice and Its Implications. Neuron 87, 451–462.

Polania R, Moisa M, Opitz A, Grueschow M, Ruff CC (2015) The precision of value-based choices depends causally on fronto-parietal phase coupling. Nat Commun 6:8090.

2) Related to the point above, the reported correlation in Figure 4F (i.e., N2 amount*delta activity and drift rate) appears to be highly exploratory and speculative. With four model parameters and many possible combinations of sleep stage durations and oscillatory activity, it is highly probable that this is a false positive. Did the authors also correct for multiple comparisons here, as in the other analyses? I would strongly suggest omitting this part or at least strongly attenuating the conclusions made from this analysis. For me, the more important point here is that the model parameters are not sensitive for cueing benefits.

See response above to reviewer 1 comment 3.

3) The authors should give some more details on the reactivation itself: How many cues were delivered on average? How many cues were delivered in N2 vs N3 vs in other sleep stages? How does number of cues relate to changes in choice preferences? Was the same snack name presented 10 times in a row or intermixed with other snack names? Please specify.

We thank the reviewer for urging us to clarify information related to the TMR procedure used in the study. First, for each subject, 8 different snack names were presented in a randomized order. To increase the chance of triggering memory replay in the sleep group, each snack name was repeated for exactly 10 times in a row before presenting the next snack item. Because both the number of snack items presented during the stimulation session and the number of repetition for each snack were fixed, we are not able to investigate how different presentation schemes would result in differential influences on choice preferences.

Second, no verbal cues were steadily delivered during sleep stages other than N2. Following previous studies (Oudiette et al., 2013; Schreiner and Rasch, 2015), verbal cueing started 2 min after a sleep group subject displayed stable N2 sleep for the first time and immediately paused when EEG recordings showed any signs of micro-arousal, awakening, or changing of sleep stages. On average, the delivery of verbal cues during nap was interrupted 0.75 ± 0.58 times. Future studies with varying numbers of cue presentation and involving stimulation at different sleep stages will be needed to provide a more thorough understanding of how selectively memory reactivation plays a role in affecting choice preferences. We have now clarified this information and included the relevant experimental details in Materials and methods.

4) Also, please give more details on the wake control group. What did this group do while listening to the stimuli? Did they perform on a different task or did they activity attend the presented stimuli? Was EEG recorded during that time? If yes, please report also the results.

See above response to reviewer 1 comment 2.

5) In general, the authors also need to make clear that they did not find any general effect of sleep on choice preferences in their study. Mechanistically, this is an important point. In the sleep and memory field, it is assumed that memories are spontaneously reactivated during sleep, and that TMR enhances this spontaneous mechanism. If choice preference depends on the degree of reactivation during sleep, then the change in WTP should be generally stronger after sleep as compared to wakefulness. (However, some results also in the sleep/memory domain suggest that general sleep associated benefits for memory are only visible after longer sleep periods, whereas TMR benefits are detectable already after short naps, see Diekelmann et al., 2012). Thus, please discuss what general effects of sleep / spontaneous reactivations during sleep you would expect on choice preference and why. In other words, does memory consolidation during sleep generally affect decisions / preferences, or is TMR simply a way of (artificially) biasing a process that does not depend on sleep itself.

We thank the reviewer for this excellent comment regarding the underlying mechanism for preference enhancement. We have now made it clearer that in our experiment daytime nap is not associated with an overall preference change, and have discussed this finding in light of the active memory consolidation model in the discussion, replicated below.

“Our results also raise intriguing questions regarding whether the preference enhancement is specific to cued memory reactivation, or hold more generally for both cued and spontaneous reactivation during sleep. […] Future investigations using night sleep or animal models will be needed to assess whether and how spontaneous memory reactivation affects the storage of reward-related information and to determine whether sleep plays a general role in modulating goal-directed behavior through tuning reward evaluation.”